# Identification of Deoxynivalenol and Degradation Products during Maize Germ Oil Refining Process

**DOI:** 10.3390/foods11121720

**Published:** 2022-06-13

**Authors:** Yuqian Guo, Tianying Lu, Jiacheng Shi, Xiaoyang Li, Kesheng Wu, Yonghua Xiong

**Affiliations:** 1State Key Laboratory of Food Science and Technology, Nanchang University, Nanchang 330047, China; yuqianguonuc@163.com (Y.G.); tianyingluai@163.com (T.L.); 2Beijing Center for Disease Prevention and Control, Beijing 100013, China; shijiachen@126.com; 3School of Food Science and Technology, Nanchang University, Nanchang 330047, China; 4Jiangxi Institute of Veterinary Drug and Feedstuff Control, Nanchang 330096, China; wufei56@126.com

**Keywords:** deoxynivalenol, maize germ oil, refining, degradation products

## Abstract

Deoxynivalenol (DON) contamination in germs and germ oil is posing a serious threat to food and feed security. However, the transformation pathway, the distribution of DON, and its degradation products in edible oil refining have not yet been reported in detail. In this work, we systematically explored the variation of DON in maize germ oil during refining and demonstrated that the DON in germ oil can be effectively removed by refining, during which a part of DON was transferred to the wastes, and another section of DON was degraded during degumming and alkali refining. Moreover, the DON degradation product was identified to be norDON B by using the ultraviolet absorption spectrum, high-performance liquid chromatography (HPLC), ultra-high-performance liquid chromatography-quadrupole time-of-flight mass spectrometry (UPLC-Q-TOF MS), and nuclear magnetic resonance (NMR) methods, and the degradation product was found to be distributed in waste products during oil refining. This study provides a scientific basis and useful reference for the production of non-mycotoxins edible oil by traditional refining.

## 1. Introduction

Most of the crops and grains in the world are subjected to harvesting, sorting, storing, and processing into various grain and oil products for human consumption. Vegetable oil, particularly germ oil, is an important grain product which provides rich nutrition to human health because it contains a high content of unsaturated fatty acid [1,2,3]. Germ oil can also dissolve cholesterol that is accumulated in the blood. However, the oil seeds are easily infected by mycotoxins such as aflatoxin B_1_ (AFB_1_), zearalenone (ZEN), and deoxynivalenol (DON) in the field or during storage [4,5,6]. Therefore, mycotoxin contamination in vegetable oil has become a serious food safety concern [7]. Among these mycotoxins, DON is a widely distributed harmful mycotoxin produced by the Fusarium species. According to a study, the detection rate and average contamination of DON in wheat germs are 60% and 111 µg/kg, compared to 40% and 41 µg/kg in wheat germ oil, respectively [8]. DON is harmful to human health, causing symptoms such as nausea, fever, headaches, and vomiting [9]. The cytotoxicity of DON primarily comes from the C12/C13 epoxy group of the molecule, which can bind to the eukaryotic ribosome and dissociate the ribosome complex, thereby preventing the synthesis of peptides and inhibiting the synthesis of protein [10,11,12]. It is well known that grains and their products are susceptible to DON contamination, and the relevant standards of many countries stipulate the DON limit. Therefore, edible oil processed from contaminated grains has the risk of DON contamination. The development of an efficient detoxification method is crucial to ensuring vegetable oil safety and to avoiding significant economic losses. In the past decades, strategies such as ultraviolet radiation, photocatalytic degradation, and cold plasma have been developed and applied for mycotoxin detoxification in vegetable oil [13,14,15,16,17,18,19]. However, such technologies require special equipment and high energy.

To our knowledge, vegetable oils extracted from oil crops must be refined to reduce or remove the natural taste, color, and free fatty acids [2,20]. In addition, oil refining has been reported as a useful strategy to reduce the AFB_1_ and ZEN contents in vegetable oil. During refining, mycotoxins can be transformed into low-toxicity or even non-toxic degradation products. Traditional oil refining primarily includes hydration degumming, alkali refining and deacidification, and adsorption decolorization that could remove mycotoxins to varying degrees [21,22,23]. Over the past few decades, key developments have been made in the research on edible oil refining for mycotoxin removal [24]. Such studies have primarily focused on the optimization of refining to acquire the optimal detoxification conditions [25]. For example, Ji et al. reported the optimum detoxifying conditions of AFB_1_ in peanut oil with alkali refining and evaluated the safety of peanut oil after being refined with alkali, indicating that alkali refining is an effective method for removing AFB_1_ in peanut oil [26]. The optimal detoxification conditions of AFB_1_ can be used as a guide for the production of non-mycotoxins oil. Moreover, Ma et al. researched the degradation mechanism of ZEN in the alkali refining of germ oil [27]. They found that the amount of ZEN removed during neutralization was more than that of other refining steps, such as degumming and decolorization. The above studies provide an important basis and reference for developing effective quality control methods to remove mycotoxins in edible oils. To our knowledge, no report has been conducted on the transformation of DON and the distribution of transformed products during the refining of vegetable oil.

In this work, we investigated the elimination of DON and disclosed the transformation law of DON in the course of maize germ oil processing. By using HPLC-MS-MS, UPLC-Q-TOF-MS, and NMR, we systematically tracked the changes in DON content throughout the stages of germ oil refining and identified the degradation products. Moreover, we proposed a potential mechanism for the transfer and degradation of DON in maize germ oil refining and further analyzed the toxicity of DON degradation products. The elimination mechanism of DON in germ oil refining provides a useful reference for the production of non-mycotoxins in edible oil.

## 2. Materials and Methods

### 2.1. Reagents and Materials

The DON and ^13^C-DON standards used in this work were purchased from Beijing Meizheng Biotechnology Co., Ltd. (Beijing, China). HPLC-grade methyl alcohol and acetonitrile were purchased from Tedia Company Inc. (Lowa, OH, USA) and used after filtration through a 0.22 µm organic membrane. Deionized water was purified and prepared using a Milli-Q system (Millipore, Milford, MA, USA). All of the solvents for the cell viability assay were purchased from Thermo Fisher Scientific (Waltham, CA, USA). All other solvents and chemical reagents were of analytical grade. The maize germ and household squeezer were purchased from a local market.

### 2.2. Simulation of Laboratory-Scale Maize Germ Oil Refining

Laboratory-scale maize germ oil refining (including degumming, deacidification, and bleaching) was conducted on the basis of the method reported by Liu et al. [28], with some modifications. The flow diagram of the experimental simulation of germ oil refining is shown in Figure 1. The sample testing was done in the same batch of germ oil refining. To continuously monitor the succession of DON in corn germ oil processing, samples were collected before and after each refining step for the HPLC and mass spectrometry analysis. In brief, crude oil was initially obtained by pressing the maize germ. Then, 100 g of crude oil (S1) was degummed by adding 0.1% *w*/*v* phosphoric acid and 5% *w*/*v* hot water (~75 °C) with vigorous stirring for 30 min at 75 °C. Colloidal impurities (S3) were centrifuged at 4000 rpm/min for 20 min. The degummed oil (S2) was neutralized by adding 10% *w*/*v* NaOH solution at 55 °C. After stirring for 30 min, the soap stock (S5) was formed and centrifuged at 4000 rpm/min for 20 min. The neutralized oil was washed two times with water (25 wt% of oil) (S6) and then heated at 95 °C. The deacidified oil (S4) was bleached with 1% *w*/*w* of the mixed decolorizer (5% activated carbon and 95% activated clay) at 90 °C for 30 min. The mixed decolorizer (S8) was removed by centrifugation to obtain the decolorated oil (S7). During the laboratory-scale peanut oil refining, the waste products of each stage and germ oil with different refining degrees were collected and stored at 4 °C before analysis.

### 2.3. Sample Extraction and Determination

The extraction and determination of DON in germ oil samples and the waste products of each stage during the refining process were performed in accordance with the national standards (GB 5009.111—2016) in China. In brief, 5 g of maize germ oil (accurate to 0.01 g) and 400 μL of the isotope internal standard ^13^C-DON solution (1 μg/mL, diluted with acetonitrile) were added to a 50 mL centrifuge tube. After shaking it vigorously and letting it stand for 30 min, 20 mL of the extractant (acetonitrile:water = 84:16, *v*/*v*) was added in the reaction mixture. After mixing and shaking vigorously for 20 min, the mixture was centrifuged at 10,000 rpm/min for 5 min and the supernatant was transferred to a 50 mL tube. The DON in the waste products of each stage during refining was extracted by using the same procedure. The DON content in the wash water of S6 was determined after diluting (0.84 mL of acetonitrile was added to 0.16 mL of wash water). All the samples were filtered through a 0.22 µm organic ultra-filter membrane before analysis. The DON determination was performed by HPLC-MS-MS and analyzed in the multiple-reaction monitoring scan mode (MRM). The DON-specific and ^13^C-DON-specific MS–MS data are given in Appendix A. In addition, the limits of detection (LOD) and the limits of quantification (LOQ) were 1.0 and 5.0 μg/kg, respectively.

### 2.4. HPLC Analysis

The DON residues in the water or the extracting solution of germ oil were analyzed by HPLC. The methods and conditions of the HPLC analysis refer to the previously reported methods, with some modifications [27]. Typically, the sample (20 µL) was injected to an Agilent C18 column (250 mm × 4.6 mm; particle size, 1.9 µm) at 40 °C. The mobile phase consisting of methanol–water (30:70 [*v*/*v*]) was used at a flow rate of 0.6 mL/min, and the injection duration was 15 min at 218 nm. The linear range of this method is 10–1000 ng/mL, and the LOD is 5 ng/mL.

### 2.5. HPLC–QToF MS Analysis of Degradation Products

The methods and conditions of the HPLC–Quadrupole Time-of-Flight MS–MS analysis refer to previously reported methods, with some modifications [29]. Briefly, Chromatographic evaluations were performed on an Agilent 1260 series HPLC system equipped with a binary pump, a micro degasser, an autosampler, and a temperature-controlled column compartment. Chromatographic separations were performed on a C18 column (100 mm × 2.1 mm, with a 1.8 µm particle size). The mobile phase consisting of A:B (20:80, *v*/*v*) was used at a 0.6 mL/min flow rate at 35 °C; the mobile phase A was 0.1% formic acid water solution and the mobile phase B was acetonitrile. The sample volume was 20 μL. The operating parameters were as follows: drying gas (N_2_) flow rate, 10 L/min; temperature, 325 °C; nebulizer, capillary voltage, 3500 V; fragmentor 125 V. The samples were analyzed in the negative ion mode, and the MS data were collected in the *m*/*z* range of 50–1000. To ensure the desired mass accuracy of the collected ions, the reference masses Purine and HP-0921 were utilized for internal mass calibration. The extracts were analyzed in a mass range of 50–1000 *m*/*z*, and the isolation width was set to medium (~4 *m*/*z*) [29].

### 2.6. NMR Spectroscopy Analysis of Degradation Products

Before the NMR analysis, the degradation products of the DON in the water solution were isolated and purified by preparative HPLC. Chromatographic separations were implemented on a C18 column (300 mm × 30 mm; 5 µm) at 40 °C. The mobile phase consisting of A/B (20/80, *v*/*v*) was used at a flow rate of 10 mL/min; the mobile phase A was water with 0.1% formic acid and the mobile phase B was methanol. The injection duration was 15 min. The wavelength of the detector was set at 218 nm. The fractions were collected with a distillate tube from 4 to 5.5 min. The collected fractions were subjected to rotary evaporation and then freeze drying to obtain a solid powder. After the purification of the reaction products by preparative HPLC, ^1^H NMR, ^13^C NMR, ^1^H–^1^H COSY, and ^1^H–^13^C HSQC analyses were conducted on a 500 MHz NMR spectrometer at 300 ± 0.1 K. The sample analysis methods and conditions refer to previously reported methods, with some modifications [27]. A total of 20 mg of the product was dissolved in 500 μL of 50% (*v*/*v*) D_2_O (99.8% D%) H_2_O. The solution was filtered and then transferred into a tube for analysis. All the NMR data were recorded and analyzed using Topspin 3.5.7 software provided by Bruker.

### 2.7. Cytotoxicity Evaluation of Degradation Products

A cell viability assay was used to research the cytotoxicity of the DON and the products. Human colon adenocarcinoma (Caco-2) cells were used in this study [9]. Caco-2 was cultured in Minimum Essential Medium Eagle (Corning, 10-010-CV) supplemented with 10% fetal bovine serum (Gibco, 10099141C), penicillin (100 U/mL), and streptomycin (Gibco, 15140163) (100 mg/mL) and incubated at pH 7.4, 5% CO_2_ at 37 °C, and in a 95% air atmosphere. The medium was changed every 2 days. The cytotoxic effect of the DON or norDON B on the Caco-2 cells was determined on the basis of the assessment of cell viability using 2-(2-methoxy-4-nitrophenyl)-3-(4-nitrophenyl)-5-(2,4-disulfophenyl)-2H-tetrazolium and monosodium salt (WST-8) with a Cell Counting Kit-8 (CCK-8) assay (Beyotime, China). DON and norDON B were dissolved in DMSO and stored at −20 °C before dilution in cell culture media. The control cells were treated with DMSO, and the final concentration of DMSO in the control and exposure groups was 0.1% solvents [30]. The Caco-2 cells (10^4^ cells/well) were cultured in 96-well plates with 100 μL of a fresh medium. After incubation in a series of concentrations of DON or norDON B with Caco-2 cells for 24 h, the medium was replaced, containing serial dilutions (serial dilution factor = 2) of DON or norDON B. Subsequently, 100 μL of a cell suspension was incubated with 10 μL of the CCK-8 solution in 5% CO_2_ atmosphere at 37 °C for 2 h. The absorbance of the cell suspension was measured at an absorbance at 450 nm.

## 3. Results and Discussion

### 3.1. Fate of DON during Maize Germ Oil Refining

The flow diagram of the experimental simulation of maize germ oil refining is shown in Figure 1. The traditional germ oil processing adopts a continuous refining process with fast circulation, including degumming, deacidification, and decolorization, which can achieve the best refining effect. The DON-free crude oil confirmed by the HPLC-MS/MS method served as the basal material to explore the succession rule of the DON in edible oil refining.

Firstly, the dynamic change of the DON in each refining step was first explored by treating the DON-spiked crude oil (1000 μg/kg), degummed oil (1000 μg/kg), and deacidified oil (1000 μg/kg) with degumming, deacidification, and decolorization for 0 to 30 min, respectively, to achieve the maximum removal efficiency. The DON concentrations of germ oils in different refining stages and processing times were determined by HPLC. As shown in Appendix A, during degumming, the DON drastically reduced from 0 to 10 min and continuously decreased until reaching a plateau at 40% residual rates of DON in the degummed oil after treatment for 15 min. During deacidification, the DON sharply decreased by approximately 80% after 20 min until reaching the balance, analogously. However, during the decoloration, the mixed decolorizers can only approximately remove 20% of toxins. These results indicated that DON removal can achieve a balance during traditional refining. Therefore, we further investigated the transformation law of DON in the current refining process. The distribution of DON in germ oil and its waste products throughout the whole processing chain was analyzed. Analogously, 1000 μg/kg of DON was added to the DON-free crude oil; then, the spiked maize germ oil was continuously refined, including degumming, deacidification, and decoloration. As shown in Figure 2a,b and Appendix A, the residual amounts of DON in the degummed oil (S2), deacidified oil (S4), and decolorized oil (S7) were 396.8, 24.5, and 0.0 ng/g, respectively. The DON levels in the S2, S4, and S7 were significantly lower than those in the S1 (*p* < 0.0001), and the DON levels in the S4 were significantly lower than those in the S7 (*p* < 0.05). This indicated that the traditional refining process can completely remove DON in maize germ oil. Notably, more than half of the DON (50.8%) was transferred to the colloidal precipitates (S3), and the total amount of DON in the S2 and S3 (85.5%) was less than that in the crude oil (S1, 100%). Analogously, the sum of the amount of DON in the S4, S5, and S6 (12.9%) was less than that in the S2 (34.7%). Based on the abovementioned results, we hypothesized that, apart from being transferred to waste products, including colloidal precipitates, saponins, and wash water, the DON was also degraded during refining. Therefore, the identification of DON degradation products and the tracing of their distribution in germ oil and waste products are particularly critical to ensuring the safety of germ oil.

### 3.2. Identification of DON Degradation Products

The abovementioned results suggested that the traditional refining process for germ oil could effectively remove and degrade DON. We further surveyed the structure and distribution of the DON degradation products during the germ oil refining to investigate whether the refined oil is safe or not. Considering the complexity of the food matrix components (the total ion chromatogram of the germ oil extract is provided in Appendix A), we first explored the DON degradation product in an aqueous solution by adding a citric acid solution or sodium hydroxide to the aqueous DON solution for simulation refinement. Briefly, 100 µL of 0.1% *w*/*w* citric acid or 10% *w*/*v* NaOH solution was added to the DON solution at 75 °C; then, the mixture was incubated for 15 min to simulate degumming or deacidification. Subsequently, the reaction solution was analyzed by using an ultraviolet–visible spectrophotometer and HPLC. As shown in Figure 3a, compared with that of the DON, the maximum absorption wavelength in the UV–vis spectra of the DON variations showed a remarkable blue shift from 218 nm to 210 nm or 206 nm after acid or alkali treatment. In the HPLC spectra, the DON chromatographic peak (retention time of 8.64 min) remarkably decreased after alkali treatment, and a new peak centered at the retention time of 4.27 min may be attributed to the continuous increase in the degradation product with the increase in the DON initial concentration (Figure 3b). However, the DON was barely diminished after the acid treatment in the aqueous solution (Appendix A), which was not in agreement with the results of the germ oil processing (Figure 2b and Appendix A), indicating that complex interactions may be observed between DON and food components during degumming.

The structure of the DON degradation product during deacidification was further investigated by Q-TOF MS-MS. On the basis of the obtained accurate mass of the parent ions and fragments, the possible structures of the DON degradation product and its fragments are shown in Figure 4. The ESI spectrum showed that the *m*/*z* of the protonated molecular ion [M–H^+^]^−^ of the DON was 295.1157 at the negative ion mode (Appendix A). The *m*/*z* of the protonated molecular ion [M–H^+^]^−^ of the DON degradation product was 265.1125 (C_14_H_17_O_5_), whereas the *m*/*z* of its secondary fragments were 247.1010 (C_14_H_15_O_3_), 223.0990 (C_12_H_15_O_4_), 193.1089 (C_11_H_13_O_3_), 175.1126 (C_11_H_11_O_2_), and 163.0777 (C_10_H_11_O_2_) (Figure 4b). Furthermore, the degradation product was purified by preparative HPLC and analyzed by NMR. The purified DON degradation product was confirmed by using the ^1^H NMR, ^13^C NMR, ^1^H–^1^H COSY, and ^1^H–^13^C HSQC spectra (Appendix A and Appendix A), and the results showed that its structure was consistent with the thermal degradation product of the DON reported previously (norDON B) [31].

To further explore the degradation products in the DON corn germ oil refining process, we performed HPLC and Q-TOF-MS spectrometry on the extracts of germ oil and the waste products of all stages in Figure 1. As shown in Figure 5, the DON content (grey shadow) decreased gradually in the germ oil (Figure 5a–d), and the section of the DON was transferred to colloidal impurities (Figure 5e), the saponin (Figure 5f), and the decolorizer (Figure 5h). Notably, the whereabouts of norDON B during the germ oil processing was traced by UPLC-Q-TOF-MS, showing that norDON B was distributed in the saponin and wash water (Appendix A). The results of the UPLC-Q-TOF-MS were in agreement with those of the HPLC (Figure 5f,g). However, norDON B was not detected in the refined oil. The abovementioned results indicated that DON in germ oil can be effectively removed by using the traditional refining process, whereas the DON and its degradation products were distributed in waste products after refining. This result was of great significance for guiding edible oil processing.

### 3.3. Possible Conversion Mechanism and Cytotoxicity of norDON B

norDON B corresponded to (2R,3R,3aR,8bR)-3a-(Hydroxymethyl)-6,8b-dimethyl-2,3,3a,8b-tetrahydro-1H-benz [b]cyclopenta[d]furan-2,3,5-triol. A possible mechanism for the rearrangement of DON to norDON B is shown in Appendix A, which involves retro-aldol rearrangement and enolate anion and then proceeds to norDON-B. The safety of the degradation product of the DON obtained from the oil refining waste products must be considered. The reaction product was purified by preparative HPLC to evaluate the toxicity of norDON B. The cytotoxic effect of the DON or norDON B on Caco-2 cells was determined on the basis of the assessment of cell viability using 2-(2-methoxy-4-nitrophenyl)-3-(4-nitrophenyl)-5-(2,4-disulfophenyl)-2H-tetrazolium and monosodium salt (WST-8) with a CCK-8 assay. As shown in Figure 6, the half maximal inhibitory concentrations (IC_50_) of the DON and norDON B were 2.755 and 2093 ng/mL, indicating that the cytotoxicity of the norDON B significantly decreased compared with that of the DON. However, norDON B still has weak toxicity; thus, the safety of the by-products obtained from the oil refining still needs to be concerned.

## 4. Conclusions

In this study, the rules of DON transfer and degradation during maize germ oil refining were investigated for the first time. We found and demonstrated that the DON in maize germ oil can be effectively eliminated by using the traditional refining process, in which a part of the DON was directly transferred into the wastes (colloidal impurities, saponin, and wash water) during degumming and alkali refining, and another section of the DON was degraded to norDON B during alkali refining. Moreover, we confirmed that the degradation product of norDON B was distributed in the waste products. These results provide a scientific basis for developing quality and process control methods to effectively control the degree of pollution of mycotoxins in vegetable oils.

## Figures and Tables

**Figure 1 foods-11-01720-f001:**
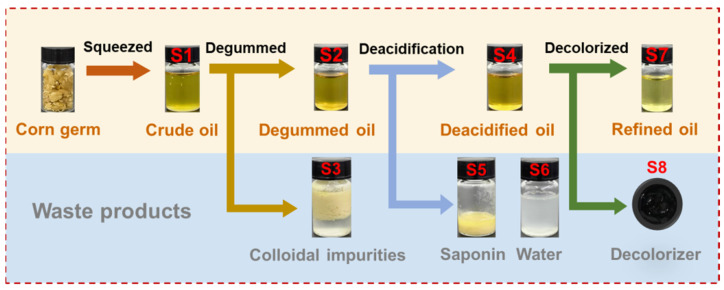
The flow diagram of simulating the maize germ oil refining process in the laboratory.

**Figure 2 foods-11-01720-f002:**
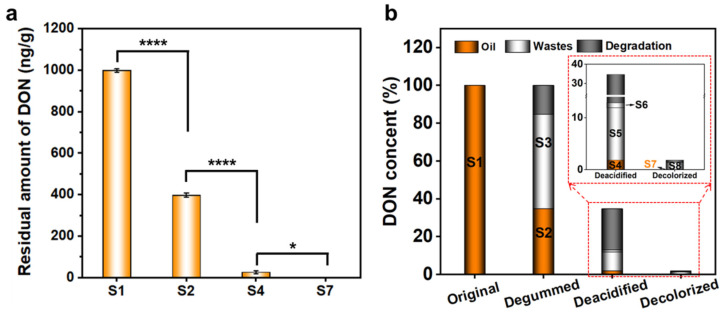
The transfer law of DON in the germ oil refining process. (**a**) The residual amount of DON in the crude oil (S1), degumming oil (S2), deacidified oil (S4), and decolorized oil (S7). The data are expressed as the mean ± SEM (standard error of mean). * *p* < 0.05, **** *p* < 0.0001 (**b**) The residual rates of DON in the germ oil continuous refining process. Each data point and error bar represent the mean and standard deviation from at least three independent measurements.

**Figure 3 foods-11-01720-f003:**
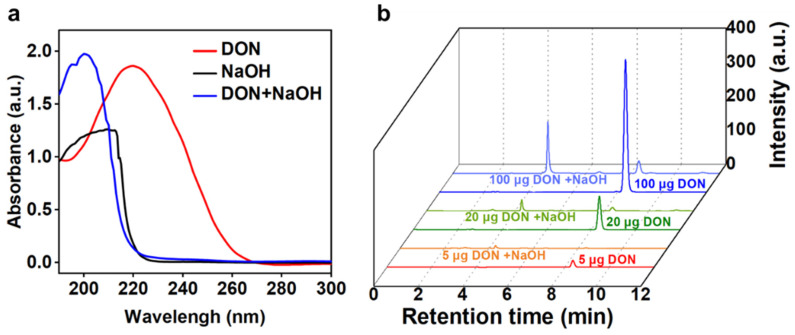
Characterizations of the DON variations in the simulated deacidification process in the aqueous solution. (**a**) UV-vis absorption spectra of the DON or the solution of alkali treatment. (**b**) HPLC chromatogram spectra of the different amounts DON and the solutions after alkali treatment.

**Figure 4 foods-11-01720-f004:**
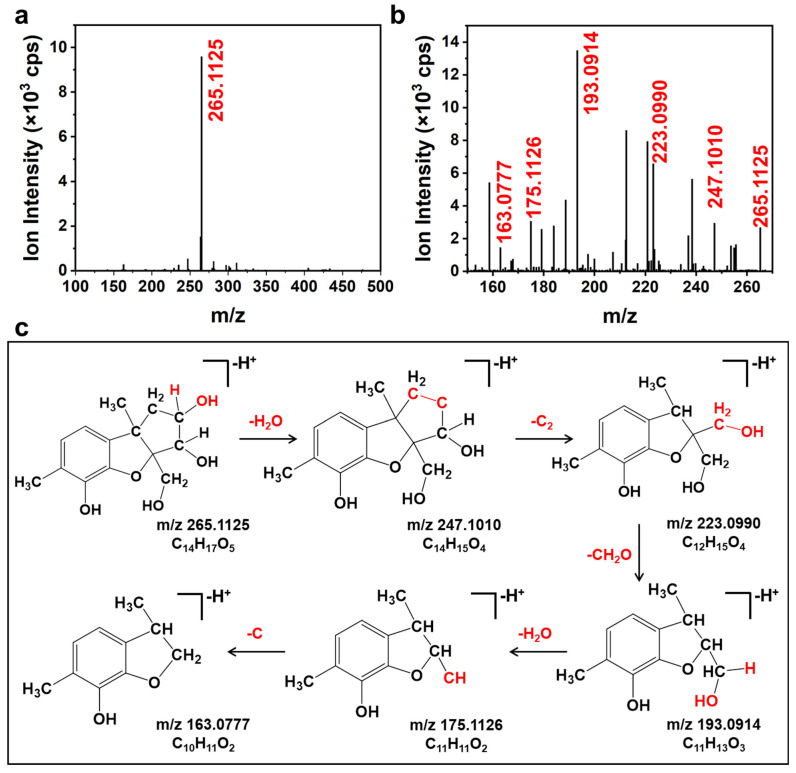
TOF MS spectra (**a**), TOF MS-MS spectra, (**b**) and possible fragments (**c**) of norDON B obtained by alkali treatment.

**Figure 5 foods-11-01720-f005:**
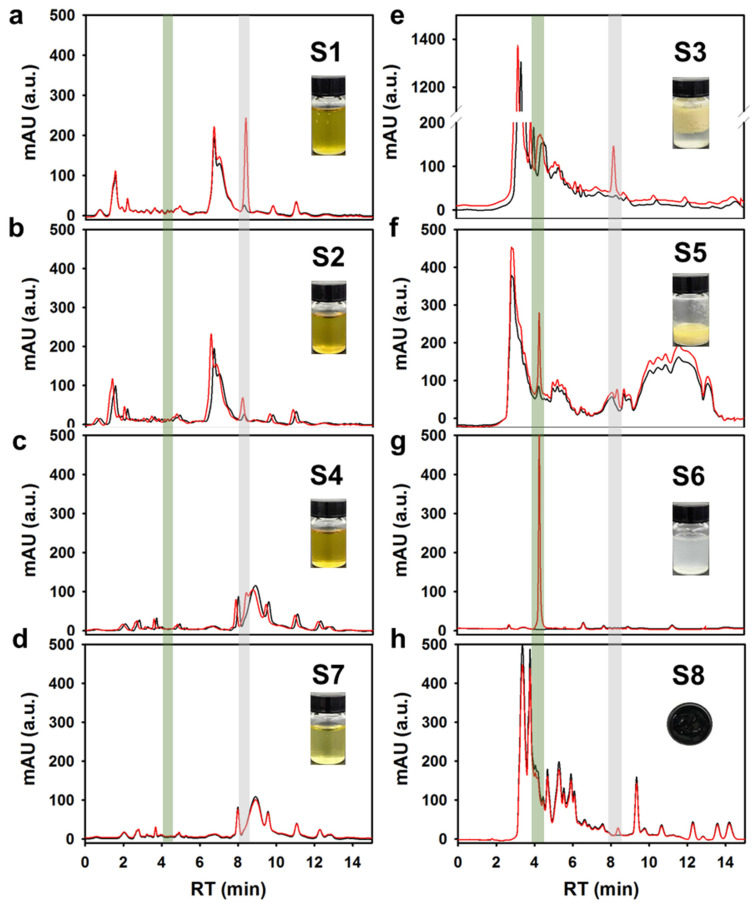
HPLC chromatogram spectra. The germ oil with different refined degrees and wastes were extracted by an acetonitrile aqueous solution (84%, *v*/*v*) with (red line) or without (black line) DON, including crude oil (**a**), degummed oil (**b**), deacidified oil (**c**), decoloration oil (**d**), colloidal impurities (**e**), soap stock (**f**), wash water (**g**), and clay (**h**).

**Figure 6 foods-11-01720-f006:**
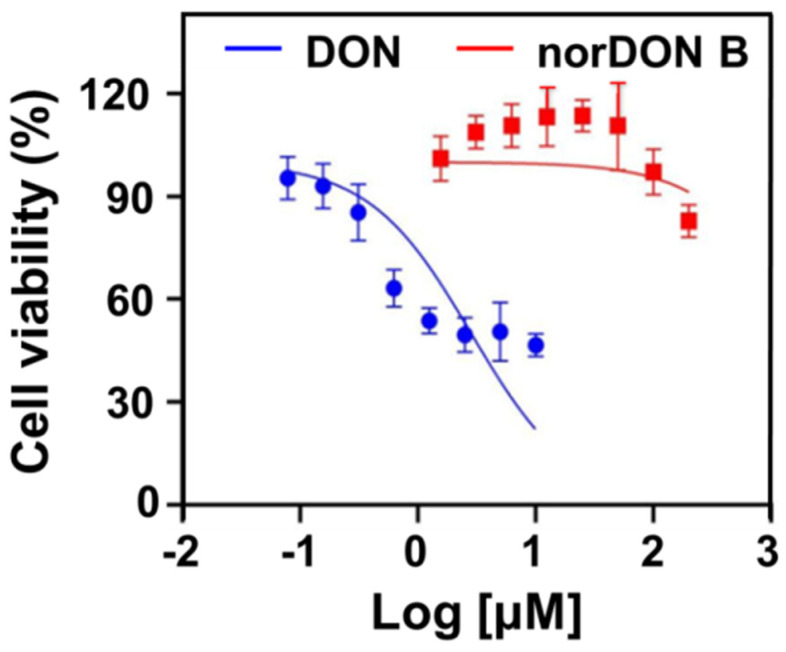
Cytotoxicity evaluation of the DON (blue) and norDON B (red).

## Data Availability

The data presented in this study are available on request from the corresponding author.

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
