# Peer review of "Identification of Deoxynivalenol and Degradation Products during Maize Germ Oil Refining Process"

_foods, 2022, doi:10.3390/foods11121720_

Round 1
Reviewer 1 Report
Dear Authors,
The article presents a very important topic. the content has been prepared appropriately and the results are comprehensive. However, I have a few comments:
1. Was the test performed on one batch of oil? If so, how can it be concluded that to one degree or another the refining processes contribute to the purification of the oil from DON?
2. Page 1 line 39 - what are the limits in other countries?
Author Response
Response to Reviewer 1 Comments
The article presents a very important topic. The content has been prepared appropriately and the results are comprehensive. However, I have a few comments:
Point 1: Was the test performed on one batch of oil? If so, how can it be concluded that to one degree or another the refining processes contribute to the purification of the oil from DON?
Response 1: Thanks for your suggestions and positive comments. Indeed, the sample testing was done in the same batch of germ oil refining. To continuously monitor the succession of DON in corn germ oil processing, samples were collected before and after each refining step for HPLC and mass spectrometry analysis, as shown in Figure 1. Additionally, in order to get the most realistic results, we carried out three batches of repeated experiments and got consistent results.
The corresponding changes were added in the revised paper in page 2 of line 92 “The sample testing was done in the same batch of germ oil refining. To continuously monitor the succession of DON in corn germ oil processing, samples were collected before and after each refining step for HPLC and mass spectrometry analysis” and in page 6 of line 233 “Each data point and error bar represent the mean and standard deviation from at least three independent measurements.”
Point 2: Page 1 line 39 - what are the limits in other countries?
Response 2: Thank you for your question. Sorry, we didn't describe clearly and we modified the relevant expression to be ”limited standard of DON” in the revised manuscript. DON is one of the mycotoxins with high pollution rate. So many countries including China, the United States, and the European Union have made limited regulations on DON in cereals and their products, alcohol, soy sauce, vinegar, sauce and sauce products. However, there is basically no country limits the amount of DON in edible oil.
The corresponding changes were added in the revised paper in page 1 of line 39: “Although most countries have no limited standard of DON in vegetable oils, the risk of DON contamination in vegetable oils still exists, which cannot be ignored.”

Reviewer 2 Report
The authors investigated on the changes of DON in maize germ oil during refining. The topic is very interesting but in my opinion some improvements must be apported.
The most criticism is the lack of statistical analysis on the experimental data; furthermore, the authors must specify how many trials were carried out starting from maize germ oil. Finally, the refined oil is the one obtained after the deodorization phase. Why didn't the authors carry out the deodorization phase?
Author Response
Response to Reviewer 1 Comments
The article presents a very important topic. the content has been prepared appropriately and the results are comprehensive. However, I have a few comments:
Point 1: Was the test performed on one batch of oil? If so, how can it be concluded that to one degree or another the refining processes contribute to the purification of the oil from DON?
Response 1: Thanks for your suggestions and positive comments. Indeed, the sample testing was done in the same batch of germ oil refining. To continuously monitor the succession of DON in corn germ oil processing, samples were collected before and after each refining step for HPLC and mass spectrometry analysis, as shown in Figure 1. Additionally, in order to get the most realistic results, we carried out three batches of repeated experiments and got consistent results.
The corresponding changes were added in the revised paper in page 2 of line 92 “The sample testing was done in the same batch of germ oil refining. To continuously monitor the succession of DON in corn germ oil processing, samples were collected before and after each refining step for HPLC and mass spectrometry analysis” and in page 6 of line 233 “Each data point and error bar represent the mean and standard deviation from at least three independent measurements.”
Point 2: Page 1 line 39 - what are the limits in other countries?
Response 2: Thank you for your question. Sorry, we didn't describe clearly and we modified the relevant expression to be ”limited standard of DON” in the revised manuscript. DON is one of the mycotoxins with high pollution rate. So many countries including China, the United States, and the European Union have made limited regulations on DON in cereals and their products, alcohol, soy sauce, vinegar, sauce and sauce products. However, there is basically no country limits the amount of DON in edible oil.
The corresponding changes were added in the revised paper in page 1 of line 39: “Although most countries have no limited standard of DON in vegetable oils, the risk of DON contamination in vegetable oils still exists, which cannot be ignored.”

Round 2
Reviewer 2 Report
The paper can be accepted in the present form
Author Response
Thank you very much for your valuable comments and suggestions to further improve our manuscript.